# Obesity and the risk of developing chronic diseases in middle-aged and older adults: Findings from an Australian longitudinal population survey, 2009–2017

Syed Afroz Keramat [1,2,3]*, Khorshed Alam[2,3], Rezwanul Hasan Rana[2,4], Rupok Chowdhury [1], Fariha Farjana[1], Rubayyat Hashmi[2,3], Jeff Gow[2,5], Stuart J. H. Biddle[3]

**1** Economics Discipline, Social Science School, Khulna University, Khulna, Bangladesh, **2** School of Business, University of Southern Queensland, Toowoomba, QLD, Australia, **3** Centre for Health Research, University of Southern Queensland, Toowoomba, QLD, Australia, **4** The Centre for the Health Economy, Macquarie University, Sydney, NSW, Australia, **5** School of Accounting, Economics, and Finance, University of KwaZulu-Natal, Durban, South Africa

* syed.afroz@econ.ku.ac.bd

**Data Availability Statement:** The data used for the study were collected by the Melbourne Institute of Applied Economic and Social Research. There are

## Abstract

### Background

Overweight and obesity impose a significant health burden in Australia, predominantly the middle-aged and older adults. Studies of the association between obesity and chronic diseases are primarily based on cross-sectional data, which is insufficient to deduce a temporal relationship. Using nationally representative panel data, this study aims to investigate whether obesity is a significant risk factor for type 2 diabetes, heart diseases, asthma, arthritis, and depression in Australian middle-aged and older adults.

### Methods

Longitudinal data comprising three waves (waves 9, 13 and 17) of the Household, Income and Labour Dynamics in Australia (HILDA) survey were used in this study. This study fitted longitudinal random-effect logistic regression models to estimate the between-person differences in the association between obesity and chronic diseases.

### Results

The findings indicated that obesity was associated with a higher prevalence of chronic diseases among Australian middle-aged and older adults. Obese adults (Body Mass Index [BMI] $\geq$ 30) were at 12.76, 2.05, 1.97, 2.25, and 1.96, times of higher risks of having type 2 diabetes (OR: 12.76, CI 95%: 8.88–18.36), heart disease (OR: 2.05, CI 95%: 1.54–2.74), asthma (OR: 1.97, CI 95%: 1.49–2.62), arthritis (OR: 2.25, 95% CI: 1.90–2.68) and depression (OR: 1.96, CI 95%: 1.56–2.48), respectively, compared with healthy weight counterparts. However, the study did not find any evidence of a statistically significant association between obesity and cancer. Besides, gender stratified regression results showed that

some restrictions on accessing this data and it is not available to the public. Those interested in accessing this data should contact the Melbourne Institute of Applied Economic and Social Research, The University of Melbourne, VIC 3010, Australia (Tel: +61 3 8344 2073 or Fax: +61 3 9347 6739).

**Funding:** The authors received no specific funding for this work.

**Competing interests:** The authors have declared that no competing interests exist.

**Abbreviations:** BMI, Body Mass Index; HILDA, Household, Income and Labour Dynamics in Australia Survey; AOR, Adjusted Odds Ratios; WHO, World Health Organization.

obesity is associated with a higher likelihood of asthma (OR: 2.64, 95% CI: 1.84–3.80) among female adults, but not in the case of male adults.

## Conclusion

Excessive weight is strongly associated with a higher incidence of chronic disease in Australian middle-aged and older adults. This finding has clear public health implications. Health promotion programs and strategies would be helpful to meet the challenge of excessive weight gain and thus contribute to the prevention of chronic diseases.

## Introduction

According to the World Health Organisation (WHO), 1.9 billion adults in the world were either overweight or obese in 2016, and the prevalence of obesity has increased threefold since 1975 [1]. It is also estimated that at least 7% of deaths from all causes globally in 2015 were related to overweight or obesity [2]. In 2017–2018, 67% (12.5 million) of Australian adults were overweight or obese, increasing from 63.4% in 2014–2015. In Australia, the prevalence of severe obesity (BMI $\geq$ 35kg/m$^2$) has almost doubled between 1995 and 2014–15 [3]. A recent study also confirmed that over one in four Australian adults (26%) were obese in 2019 [4]. Overweight and obesity impose a considerable burden (both direct and indirect) in Australia. Overweight and obesity contributed 8.4% of the risk factor of the burden of diseases in Australia in 2015 [5]. Besides, there is evidence that obesity is strongly associated with a higher acquisition of disability [6]. Further, obese Australias are more likely to report poor general health and mental health [7]. Moreover, obesity has a substantial negative impact on diverse labour market outcomes, such as high absenteeism [8], increased presenteeism [9], job dissatisfaction [10], and a higher rate of job discrimination [11].

There is increasing empirical evidence that obesity triggers the likelihood of different non-communicable diseases (NCDs), such as type 2 diabetes, high blood pressure, cardiovascular disease (CVD), cancer, asthma, sleep apnea, and poor mental health [12]. An excessive gain of body weight from early childhood to adulthood is consistently associated with the risk of heart disease [13]. Obesity is also significantly related to the risk of heart disease-related morbidity and mortality [14]. Further, it is strongly associated with the incidence of type 2 diabetes [15] and depression [16]. Furthermore, the likelihood of different patterns of arthritis, such as osteoarthritis, rheumatoid arthritis, and psoriatic arthritis, is often associated with increased body weight [17]. The burden of these chronic diseases includes low quality of life, productivity loss, and increased healthcare costs [18, 19].

While the prevalence of obesity and chronic diseases is high across Australia, people from lower socioeconomic backgrounds are often disproportionately affected [20]. Although there is a clear link between obesity and chronic health conditions, the severity of the burden of risk might vary based on an individual's socioeconomic and demographic conditions as well as lifestyle characteristics. For policy-making purposes, it is crucial to understand whether obesity causes an increase in specific types of chronic disease among the poor, the elderly, and physically inactive compared to the affluent, younger and/or physically active population. Previous studies estimating the obesity and chronic disease nexus in Australia often focused on a single disease using cross-sectional survey data, which is insufficient to deduce a temporal relationship. Besides, there is a lack of emphasis on the critical confounding factors (e.g. socioeconomic and demographic) that might explain the severity of the risks of obesity for a specific

cohort of people, but not others. There is also a lack of literature that has employed nationally representative longitudinal survey data to examine the association between obesity and chronic disease burden. Longitudinal designs are essential for the understanding of the dynamics of the relationship and interdependence (e.g., the link between obesity and chronic diseases) and to better identify the influence of one factor (e.g., obesity) over the other (e.g., chronic diseases). Therefore, this study aims to fill these gaps in the literature by employing the longitudinal study design. The main objective of this study is to estimate the between-person differences in the relationship between obesity and chronic diseases in Australian adults. To the best of the authors' knowledge, no previous research has focused on the obesity and chronic disease nexus from the Australian perspective, especially for middle-aged and older adults using longitudinal data.

## Materials and methods

### Data source and sample selection

The study utilised nationally representative data from the Household, Income and Labor Dynamics in Australia (HILDA) survey. The HILDA survey was initiated in 2001 by collecting detailed information on 13,000 individuals within 7,000 households using a multistage sampling approach. Since then, the survey has gathered information on a wide range of topics: wealth, retirement, fertility, health, education, skills and abilities from members of households aged 15 years or over through a self-completed questionnaire (SCQ) and face-to-face interviews by trained interviewers. The description of the HILDA survey design is shown elsewhere [21].

Participants of this longitudinal study were selected from three waves (waves 9, 13 and 17) of the HILDA survey, and data were collected during the years 2009, 2013 and 2017, respectively. The reason behind considering these waves was that these three waves substantially capture the respondents' health and lifestyle-related characteristics. Fig 1 demonstrates the procedure of obtaining the final analytic sample. The analytical sample is restricted to adults aged 45 years and over. The inclusion criteria for the subsample analyses were no missing information on participants' Body Mass Index (BMI) and chronic diseases. This study also excludes pregnant women's data to avoid potential biases. The final analytic sample consisting of 20,538 person-year observations from 9,822 unique participants was achieved by applying inclusion and exclusion criteria.

### Outcome variable

The outcome variable of the study is self-reported chronic disease. The HILDA survey collects information on an individual's chronic disease status by asking questions, 'are you diagnosed with a serious illness?' This study considered six types of chronic diseases, including type 2 diabetes, heart disease, asthma, cancer, arthritis and depression, as the outcome variables of interest. Responses on the outcome variables were taken in binary form (0 = no, 1 = yes).

### Exposure variable

This study checks if obesity is a significant risk factor for chronic diseases among Australian middle-aged and older adults. The current study measures obesity through BMI. HILDA survey collects data on BMI using self-reported weight and height following the formula of weight (in kilograms) divided by height (in metres) square. The authors categorised BMI as underweight (<18.50), normal/healthy weight (18.50–24.99), overweight (25.00–29.99), and obese (≥ 30.00) following WHO guidelines [1]. This classification allows an assessment of how and

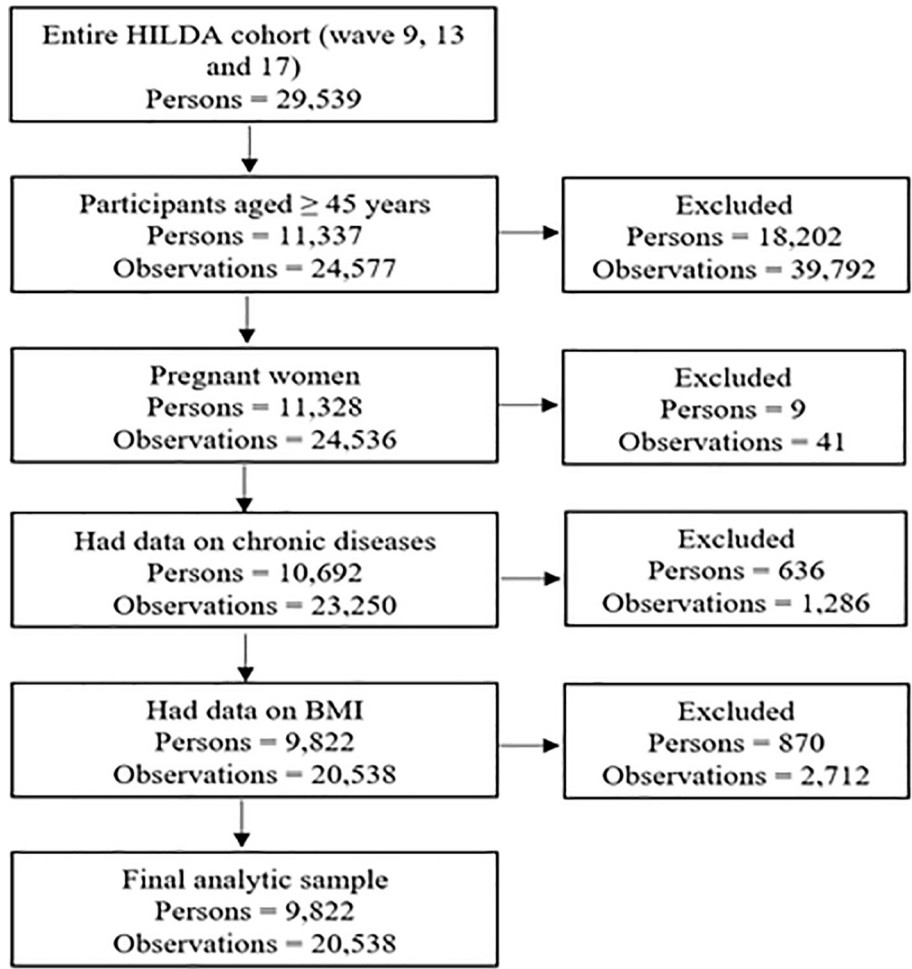

**Fig 1. Flow chart of sample selection and missing data.**

in what context underweight, overweight and obese participants are susceptible to different chronic diseases compared with their healthy weight counterparts.

## Other covariates

This study considered potential confounders following previous studies [22, 23]. One significant advantage of the HILDA survey is that it provides a considerable amount of data on the demographic characteristics of respondents, such as age, gender, income level, education, area of residence and other behavioural factors. Table 1 shows the set of the confounders with their nature and categories considered for the present study. For instance, age is categorised as middle-aged (45 to 59 years) and older adults (≥ 60 years). Other socio-demographic confounders include gender (male and female), civil status (partnered, unpartnered), education (year 12 or below, professional qualifications, and university qualifications), household yearly disposable income (expressed in quintiles), labour force status (employed, unemployed, and not in the labour force), Indigenous status (non-Indigenous, and Aboriginal/Torres Strait Islander [ATSI]), location (major city, regional city and remote areas).

Besides, three behavioural factors: smoking status, alcohol consumption and physical activity, served as the confounders. Smoking status was categorised as never smoked, ex-smoker,

**Table 1. Description of other covariates.**

| Covariates | Categories |
|---|---|
| Age | Middle-aged (45 to 59 years), and older adults ($\geq$ 60 years). |
| Gender | Male and female. |
| Civil status | Partnered (married, and never married but living with someone in a relationship), and unpartnered (separated but not divorced, divorced, widowed, and never married and not living with someone in a relationship). |
| Education | Year 12 and below (year 12, and Year 11 and below), professional qualifications (advance diploma or diploma, and certificate III or IV), and university qualifications (postgraduate—masters or doctorate, graduate diploma or certificate, bachelor or honours). |
| Household yearly disposable income quintile | Quintiles (quintile 1 [lowest] to quintile 5 [highest]). |
| Labour force status | Employed, unemployed, and not in the labour force. |
| Indigenous status | Non-Indigenous, and Aboriginal/Torres Strait Islander (ATSI) or both. |
| Location | Major city, regional city (inner and outer regional) and remote areas (remote and very remote). |
| Smoking status | never smoked, a former smoker and current smoker. |
| Alcohol consumption | Never drank, ex-drinker, only rarely to 4 days/week, and 4+ days/week. |
| Physical activity ($\geq$ 30 minutes) | Not at all to <1/week, 1–3 times/week, and $\geq$ 4 times/week. |

and current smoker. The variable alcohol consumption was classified as never drank, ex-drinker, only rarely to four days and more than four days per week. Physical activity-related information was collected by questioning how often the respondent participates in physical activity each week for at least 30 minutes. This study categorised physical activity as: not at all to less than one, 1 to 3 times, and more than three times per week.

## Estimation strategy

The authors prepared an unbalanced longitudinal data set consisting of 20,538 person-year observations by linking de-identified records of 9,822 unique adults. This study considered three distinct waves (waves 9, 13, and 17) of the HILDA survey covering the period from 2009 to 2017. Due to the longitudinal nature of the data, repeated observations on the same individual were used for subsample analyses. This study reports baseline, final wave, and pooled prevalence of obesity, six chronic diseases, socio-demographic and behavioural characteristics in the form of frequency(n) and percentages (%) with 95% confidence intervals (CI). The relationships between the exposure and other covariates with chronic diseases were first identified through bivariate analysis (test results not reported here). Statistically significant (*P*-value <0.05) variables in the bivariate analyses were then considered for the final regression model.

This study employed the longitudinal random-effects logistic regression model to capture between-person variation as the study data were derived from a longitudinal dataset (repeated measures). The outcome variables (type 2 diabetes, heart disease, asthma, cancer, arthritis and depression) are binary (whether they have a particular chronic condition or not). Therefore, this study utilised the logistic link. To ease the interpretation, this study reports regression results in the form of adjusted Odds Ratios (AOR) along with the 95% confidence interval. This study sets p-value <0.05 for the statistical significance of a variable. A variable will be considered statistically significant if the p-value for the variable is less than the significance level in the regression models. All statistical analyses were performed using Stata, version 16 (StataCorp LLC).

### Ethics approval

This study did not require ethical approval as the analysis used only de-identified existing unit record data from the HILDA survey. However, the authors completed and signed the Confidentiality Deed Poll and sent it to NCLD (ncldresearch@dss.gov.au) and ADA (ada@anu.edu.au) before the data applications' approval. Therefore, datasets analysed and/or generated during the current study are subject to the signed confidentiality deed.

## Results

Table 2 displays the characteristics of the study participants in terms of their chronic diseases, socio-demographic, and behavioural characteristics at the baseline, final, and pooled in all waves. Among the study participants, 47% were male, and 53% were female, a higher proportion (53.26%) were middle-aged, nearly two-thirds (65.53%) were unpartnered, over one-fifth (21.86%) had university qualifications, over half were employed (53.22%), primarily non-Indigenous and lived in major cities (61.96%) at the baseline. The results also show that nearly 48% of participants never smoke, 59% consume alcohol from rarely to four days per week, and 35% performed physical activities that last at least 30 minutes over three times per week (baseline wave).

Of 9,822 participants (20,538 observations), approximately 38.48% were overweight, and 28.11% were obese. The pooled prevalence of chronic conditions, such as type 2 diabetes, heart diseases, asthma, cancer, arthritis, and depression in study participants was approximately 9.01%, 8.35%, 9.96%, 5.68%, 30.64%, and 13.0%, respectively (pooled in all waves).

Fig 2 displays the overall prevalence of various chronic diseases among Australia's middle-aged and older adults at three different periods: 2009, 2013 and 2017. Fig 2 manifests that the prevalence of chronic conditions and obesity among the study population had increased from 2009 to 2017. Among all of them, depression increased sharply from 10% to 15% approximately. Incidence of type 2 diabetes, asthma, and arthritis marginally increased over the period, and the prevalence of heart diseases and cancer also increased over time. The prevalence of obesity was almost 25% in 2009, which increased to nearly 30% in less than ten years.

Fig 3 illustrates the percentage of chronic diseases among middle-aged and older adults based on their weight status. Prevalence of chronic conditions, such as type 2 diabetes (16.18%), asthma (12.99%) and arthritis (37.52%), was highest in obese people. However, underweight middle-aged and older adults are more vulnerable to heart diseases (11.76%), cancer (7.96%) and depression (19.72%). For obese people, the percentage is also noticeable, i.e. 10.27%, 5.68% and 17.11% for heart diseases, cancer and depression, respectively.

Fig 4 shows the prevalence of co-morbid conditions in middle-aged and older adults stratified by gender (pooled in all waves). It is observed that the prevalence of asthma (16.77% vs 8.44%), arthritis (44.55% vs 29.06%), and depression (19.53% vs 14.20%) are substantially higher among females than males. However, cancer (6.64% vs 4.88%), heart diseases (13.13% vs 7.89%) and type 2 diabetes (17.72% vs 14.90%) were more prevalent among males than females.

Table 3 exhibits the results obtained from the adjusted random-effect logistic regression model to investigate between-person differences in the relationship between obesity and six types of chronic diseases. The results show that the risk of having a chronic disease was more pronounced among obese adults compared with their healthy-weight counterparts. Obese people were at higher risks of suffering from type 2 diabetes (OR: 12.76, 95% CI: 8.88–18.36), heart diseases (OR: 2.05, 95% CI: 1.54–2.74), asthma (OR: 1.97, 95% CI: 1.49–2.62), and arthritis (OR: 2.25, 95% CI: 1.90–2.68) compared with their healthy-weight counterparts. It is also

**Table 2. Distribution of the analytic sample: Baseline, final and pooled across all waves (persons = 9,822; observations = 20,538).**

| Characteristics | Baseline wave (2009) | | Final wave (2017) | | Pooled in all waves (2009, 2013 & 2017) | |
|---|---|---|---|---|---|---|
| | n | % | n | % | n | % |
| **Outcome variables** | | | | | | |
| **Type 2 diabetes** | | | | | | |
| No | 4,946 | 91.56 | 7,082 | 90.82 | 18,688 | 90.99 |
| Yes | 456 | 8.44 | 716 | 9.18 | 1,850 | 9.01 |
| **Heart disease** | | | | | | |
| No | 4,981 | 92.21 | 7,120 | 91.31 | 18,824 | 91.65 |
| Yes | 421 | 7.79 | 678 | 8.69 | 1,714 | 8.35 |
| **Asthma** | | | | | | |
| No | 4,876 | 90.26 | 6,986 | 89.59 | 18,492 | 90.04 |
| Yes | 526 | 9.74 | 812 | 10.41 | 2,046 | 9.96 |
| **Cancer** | | | | | | |
| No | 5,096 | 94.34 | 7,339 | 94.11 | 19,372 | 94.32 |
| Yes | 306 | 5.66 | 459 | 5.89 | 1,166 | 5.68 |
| **Arthritis** | | | | | | |
| No | 3,747 | 69.36 | 5,406 | 69.33 | 14,246 | 69.36 |
| Yes | 1,655 | 30.64 | 2,392 | 30.67 | 6,292 | 30.64 |
| **Depression** | | | | | | |
| No | 4,835 | 89.5 | 6,633 | 85.06 | 17,869 | 87.00 |
| Yes | 567 | 10.5 | 1,165 | 14.94 | 2,669 | 13.00 |
| **Exposure and covariates** | | | | | | |
| **BMI** | | | | | | |
| Underweight | 83 | 1.54 | 95 | 1.22 | 289 | 1.41 |
| Healthy weight | 1,815 | 33.6 | 2,428 | 31.14 | 6,574 | 32.01 |
| Overweight | 2,133 | 39.49 | 2,942 | 37.73 | 7,902 | 38.48 |
| Obesity | 1,371 | 25.38 | 2,333 | 29.92 | 5,773 | 28.11 |
| **Age** | | | | | | |
| Middle-aged (45–59 years) | 2,877 | 53.26 | 3,713 | 47.61 | 10,304 | 50.17 |
| Older adults (≥ 60 years) | 2,525 | 46.74 | 4,085 | 52.39 | 10,234 | 49.83 |
| **Gender** | | | | | | |
| Male | 2,546 | 47.13 | 3,676 | 47.14 | 9,684 | 47.15 |
| Female | 2,856 | 52.87 | 4,122 | 52.86 | 10,854 | 52.85 |
| **Civil Status** | | | | | | |
| Partnered | 1,862 | 34.47 | 2,770 | 35.52 | 7,140 | 34.76 |
| Unpartnered | 3,540 | 65.53 | 5,028 | 64.48 | 13,398 | 65.24 |
| **Education** | | | | | | |
| Year 12 and below | 2,511 | 46.48 | 2,999 | 38.46 | 8,624 | 41.99 |
| Professional qualifications | 1,710 | 31.65 | 2,780 | 35.65 | 6,974 | 33.96 |
| University qualifications | 1,181 | 21.86 | 2,019 | 25.89 | 4,940 | 24.05 |
| **Household yearly disposable income quintile** | | | | | | |
| Quintile 1 (lowest) | 1,081 | 20.01 | 1,561 | 20.02 | 4,109 | 20.01 |
| Quintile 2 | 1,081 | 20.01 | 1,559 | 19.99 | 4,107 | 20.00 |
| Quintile 3 | 1,081 | 20.01 | 1,559 | 19.99 | 4,107 | 20.00 |
| Quintile 4 | 1,079 | 19.97 | 1,561 | 20.02 | 4,109 | 20.01 |
| Quintile 5 (highest) | 1,080 | 19.99 | 1,558 | 19.98 | 4,106 | 19.99 |
| **Labour force status** | | | | | | |

(*Continued*)

**Table 2.** (Continued)

| Characteristics | Baseline wave (2009) | | Final wave (2017) | | Pooled in all waves (2009, 2013 & 2017) | |
|---|---|---|---|---|---|---|
| | **n** | **%** | **n** | **%** | **n** | **%** |
| Employed | 2,875 | 53.22 | 4,006 | 51.37 | 10,665 | 51.93 |
| Unemployed | 75 | 1.39 | 122 | 1.56 | 326 | 1.59 |
| Not in the labour force | 2,452 | 45.39 | 3,670 | 47.06 | 9,547 | 46.48 |
| **Indigenous status** | | | | | | |
| Non-Indigenous | 5,317 | 98.43 | 7,653 | 98.14 | 20,181 | 98.26 |
| Aboriginal or Torres Strait Islander | 85 | 1.57 | 145 | 1.86 | 357 | 1.74 |
| **Location** | | | | | | |
| Major city | 3,347 | 61.96 | 4,885 | 62.64 | 12,865 | 62.64 |
| Regional | 1,968 | 36.43 | 2,792 | 35.8 | 7,352 | 35.80 |
| Remote | 87 | 1.61 | 121 | 1.55 | 321 | 1.56 |
| **Smoking status** | | | | | | |
| Never smoked | 2,597 | 48.07 | 3,878 | 49.73 | 10,034 | 48.86 |
| Former smoker | 2,004 | 37.10 | 2,855 | 36.61 | 7,609 | 37.04 |
| Current smoker | 801 | 14.83 | 1,065 | 13.66 | 2,895 | 14.10 |
| **Alcohol consumption** | | | | | | |
| Never drank | 562 | 10.4 | 785 | 10.07 | 2,101 | 10.23 |
| Ex-drinker | 379 | 7.02 | 759 | 9.73 | 1,788 | 8.71 |
| Only rarely to 4 days/week | 3,203 | 59.29 | 4,650 | 59.63 | 12,210 | 59.45 |
| 4+ days/week | 1,258 | 23.29 | 1,604 | 20.57 | 4,439 | 21.61 |
| **Physical activity (≥ 30 minutes)** | | | | | | |
| Not at all to <1/week | 1,502 | 27.80 | 2,473 | 31.71 | 6,121 | 29.80 |
| 1–3 times/week | 2,009 | 37.19 | 2,832 | 36.32 | 7,493 | 36.49 |
| ≥4 times/week | 1,891 | 35.01 | 2,493 | 31.97 | 6,924 | 33.71 |

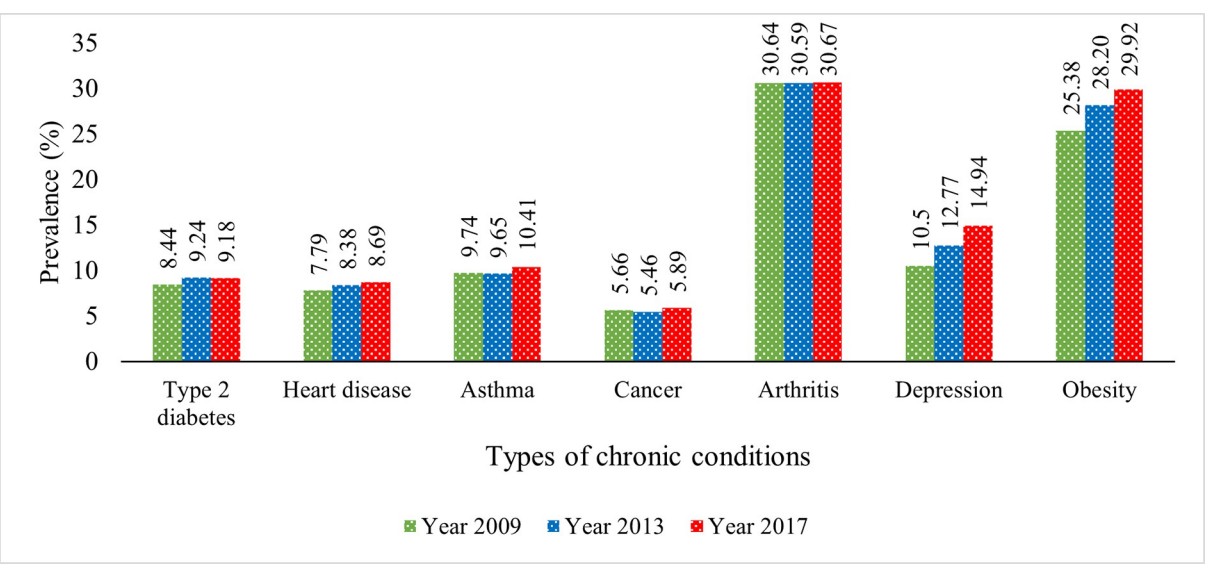

**Fig 2. Prevalence of chronic conditions among middle-aged and older adults.**

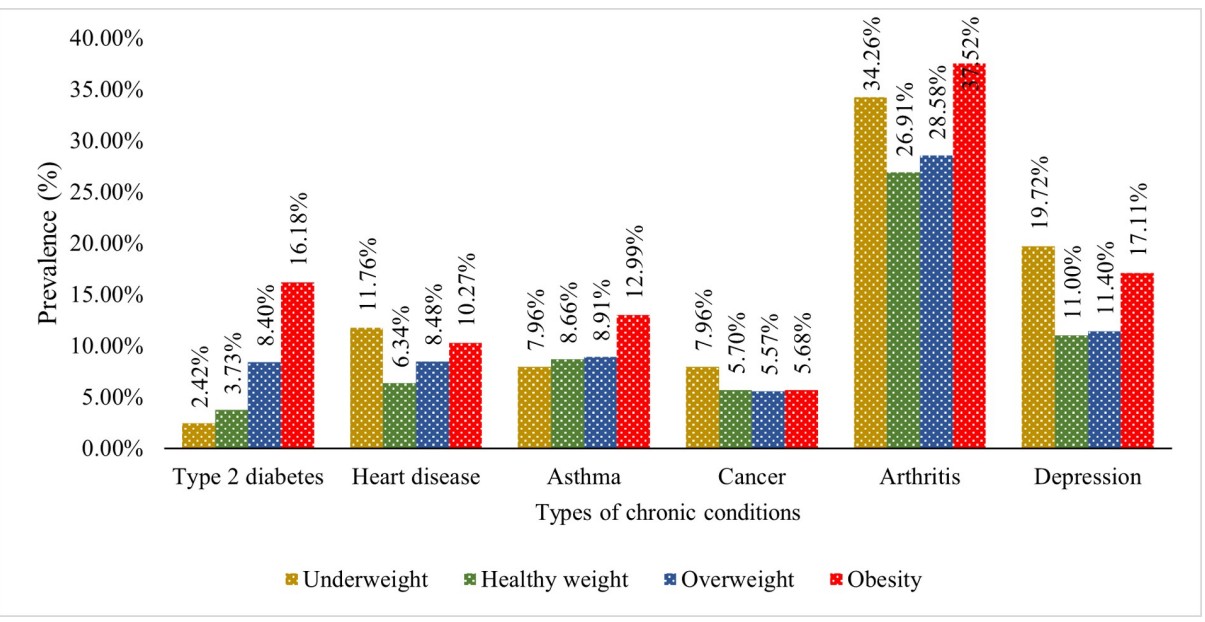

**Fig 3. Prevalence of chronic conditions among middle-aged and older adults by weight status.**

observed that obese people were at 1.96 times higher risk of suffering from depression (OR: 1.96, 95% CI: 1.56–2.48) than peers with a healthy weight.

Gender differences in the relationship between obesity and six types of chronic conditions among middle-aged and older Australian adults were reported in Table 4. The results showed that the odds of having chronic conditions, such as type 2 diabetes, heart diseases, arthritis and depression, were higher among obese adults compared to healthy weight counterparts irrespective of gender. However, the magnitudes vary with gender. For example, the risk of having

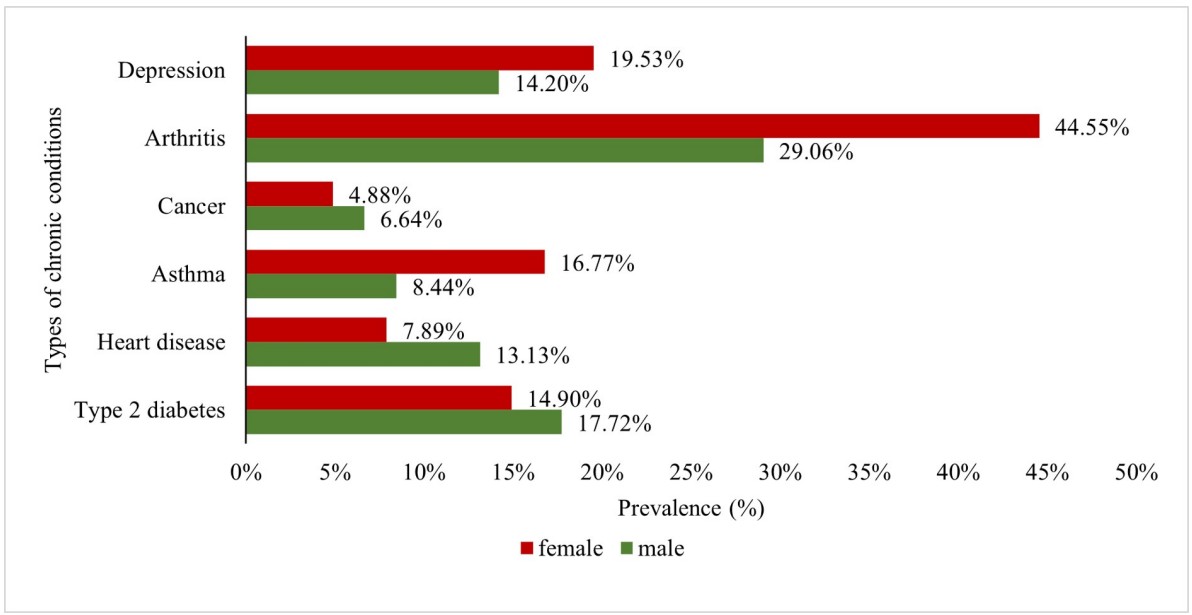

**Fig 4. Gender differences in the prevalence of the chronic conditions among obese middle-aged and older adults.**

**Table 3. Adjusted random-effect regression results for the between-person differences in chronic conditions due to obesity; 9,822 persons, 20,538 observations.**

| Variables | Model 1 | Model 2 | Model 3 | Model 4 | Model 5 | Model 6 |
|---|---|---|---|---|---|---|
| | Type 2 diabetes | Heart disease | Asthma | Cancer | Arthritis | Depression |
| | aOR (95% CI) | aOR (95% CI) | aOR (95% CI) | aOR (95% CI) | aOR (95% CI) | aOR (95% CI) |
| **BMI** | | | | | | |
| Underweight | 0.33 (0.07–1.67), 0.18 | **2.98 (1.44–6.17), 0.01** | 0.49 (0.19–1.23), 0.13 | 1.36 (0.70–2.67), 0.37 | 1.07 (0.66–1.74), 0.80 | 1.46 (0.78–2.71), 0.24 |
| Healthy weight (ref) | | | | | | |
| Overweight | **3.81 (2.71–5.36), <0.001** | **1.41 (1.09–1.82), 0.01** | 1.21 (0.94–1.56), 0.14 | 0.82 (0.66–1.01), 0.07 | **1.42 (1.22–1.64), <0.001** | **1.25 (1.01–1.54), 0.04** |
| Obesity | **12.76 (8.88–18.36), <0.001** | **2.05 (1.54–2.74), <0.001** | **1.97 (1.49–2.62), <0.001** | 0.89 (0.69–1.13), 0.33 | **2.25 (1.90–2.68), <0.001** | **1.96 (1.56–2.48), <0.001** |
| **Socio-demographic characteristics** | | | | | | |
| **Age** | | | | | | |
| Middle-aged (45–59 years) (ref) | | | | | | |
| Older adults (≥ 60 years) | **4.36 (3.23–5.89), <0.001** | **4.83 (3.60–6.48), <0.001** | 0.92 (0.72–1.19), 0.54 | **2.35 (1.87–2.96), <0.001** | **3.63 (3.12–4.21), <0.001** | **0.39 (0.31–0.48), <0.001** |
| **Gender** | | | | | | |
| Male (ref) | | | | | | |
| Female | **0.29 (0.22–0.04), <0.001** | **0.28 (0.21–0.36), <0.001** | **2.45 (1.89–3.19), <0.001** | **0.53 (0.43–0.65), <0.001** | **2.91 (2.49–3.41), <0.001** | **2.10 (1.7–2.6), <0.001** |
| **Education** | | | | | | |
| Year 12 or below (ref) | | | | | | |
| Professional qualifications | 0.89 (0.65–1.23), 0.47 | 0.88 (0.66–1.17), 0.39 | 1.07 (0.81–1.42), 0.62 | 1.25 (0.99–1.57), 0.06 | **0.80 (0.68–0.95), 0.01** | 1.19 (0.95–1.50), 0.13 |
| University qualifications | **0.62 (0.42–0.94), 0.02** | 0.92 (0.65–1.30), 0.63 | 1.08 (0.77–1.51), 0.69 | 1.05 (0.79–1.39), 0.72 | **0.61 (0.50–0.75), <0.001** | 1.02 (0.77–1.35), 0.89 |
| **Civil Status** | | | | | | |
| Partnered (ref) | | | | | | |
| Unpartnered | **0.68 (0.51–0.89), 0.01** | **0.69 (0.54–0.88), 0.01** | 0.82 (0.64–1.04), 0.10 | 0.92 (0.75–1.13), 0.45 | **0.80 (0.69–0.93), 0.01** | **0.46 (0.38–0.56), <0.001** |
| **Household yearly disposable income quintile** | | | | | | |
| Quintile 1 | **1.57 (1.04–2.38), 0.03** | 1.23 (0.86–1.75), 0.26 | **1.59 (1.12–2.27), 0.01** | 1.11 (0.81–1.51), 0.53 | **1.43 (1.16–1.77), 0.01** | **1.70 (1.27–2.29), <0.001** |
| Quintile 2 | 1.18 (0.79–1.76), 0.42 | 1.17 (0.83–1.66), 0.36 | 1.31 (0.94–1.83), 0.12 | 1.24 (0.92–1.67), 0.15 | **1.25 (1.03–1.52), 0.03** | **1.63 (1.23–2.15), 0.01** |
| Quintile 3 | 1.28 (0.87–1.89), 0.21 | 0.95 (0.67–1.35), 0.77 | 1.08 (0.78–1.49), 0.65 | 0.92 (0.68–1.25), 0.60 | 1.07 (0.89–1.29), 0.48 | **1.42 (1.09–1.86), 0.01** |
| Quintile 4 | 1.11 (0.76–1.64), 0.59 | 1.04 (0.74–1.47), 0.80 | 1.16 (0.85–1.58), 0.34 | 1.13 (0.85–1.51), 0.40 | 1.08 (0.90–1.29), 0.42 | 1.18 (0.91–1.53), 0.22 |
| Quintile 5 (ref) | | | | | | |
| **Labour force status** | | | | | | |
| Employed (ref) | | | | | | |
| Unemployed | 2.13 (0.88–5.13), 0.09 | 1.52 (0.65–3.57), 0.34 | 0.85 (0.40–1.80), 0.67 | 0.66 (0.26–1.66), 0.37 | 1.16 (0.74–1.83), 0.52 | **4.03 (2.41–6.74), <0.001** |
| Not in the labor force | **3.40 (2.48–4.66), <0.001** | **5.72 (4.18–7.85), <0.001** | **1.90 (1.45–2.49), <0.001** | **2.44 (1.91–3.11), <0.001** | **3.14 (2.68–3.68), <0.001** | **4.28 (3.41–5.37), <0.001** |
| **Indigenous status** | | | | | | |
| Non-indigenous (ref) | | | | | | |
| Aboriginal/Torres Strait Islander | **8.27 (3.37–20.34), <0.001** | **2.44 (1.08–5.52), 0.03** | 1.87 (0.82–4.30), 0.14 | 1.01 (0.47–2.12), 0.99 | 0.94 (0.55–1.60), 0.81 | 1.95 (1.00–3.81), 0.05 |

*(Continued)*

**Table 3.** (Continued)

| Variables | Model 1 | Model 2 | Model 3 | Model 4 | Model 5 | Model 6 |
|---|---|---|---|---|---|---|
| | Type 2 diabetes | Heart disease | Asthma | Cancer | Arthritis | Depression |
| | aOR (95% CI) | aOR (95% CI) | aOR (95% CI) | aOR (95% CI) | aOR (95% CI) | aOR (95% CI) |
| **Location** | | | | | | |
| Major city (ref) | | | | | | |
| Regional | 1.04 (0.78–1.37), 0.81 | 1.02 (0.80–1.30), 0.90 | 1.20 (0.94–1.54), 0.14 | 1.03 (0.84–1.26), 0.80 | **1.23 (1.07–1.43), 0.01** | 1.07 (0.88–1.31), 0.48 |
| Remote | 0.45 (0.14–1.45), 0.18 | 1.31 (0.54–3.16), 0.55 | 0.72 (0.26–2.03), 0.54 | 1.28 (0.62–2.63), 0.51 | **0.48 (0.27–0.85), 0.01** | 0.44 (0.19–1.01), 0.05 |
| **Behavioural Characteristics** | | | | | | |
| **Smoking status** | | | | | | |
| Never smoked (ref) | | | | | | |
| ex-smoker | **1.50 (1.12–2.01), 0.01** | **1.56 (1.21–2.02), 0.01** | **1.57 (1.21–2.04), 0.01** | 1.09 (0.88–1.35), 0.41 | **1.21 (1.04–1.41), 0.01** | **1.42 (1.14–1.76), 0.01** |
| Current smoker | 0.98 (0.64–1.49), 0.92 | 1.05 (0.72–1.52), 0.80 | **2.13 (1.50–30), <0.001** | 0.95 (0.69–1.29), 0.72 | 1.07 (0.86–1.32), 0.55 | **2.56 (1.95–3.36), <0.001** |
| **Alcohol consumption** | | | | | | |
| Never drink (ref) | | | | | | |
| Ex-drinker | 0.65 (0.40–1.05), 0.08 | 1.19 (0.78–1.82), 0.42 | 0.84 (0.54–1.31), 0.44 | 1.37 (0.92–2.03), 0.12 | 1.20 (0.91–1.58), 0.19 | **2.04 (1.41–2.94), <0.001** |
| Only rarely to 3 days/week | **0.44 (0.29–0.65), <0.001** | 0.71 (0.50–1.02), 0.07 | 0.72 (0.50–1.04), 0.08 | 1.18 (0.85–1.64), 0.32 | 1.17 (0.94–1.46), 0.17 | 1.12 (0.83–1.52), 0.47 |
| 3+ days/week | **0.16 (0.10–0.27), <0.001** | **0.51 (0.33–0.77), 0.01** | 0.74 (0.48–1.13), 0.17 | 1.12 (0.78–1.62), 0.54 | 1.19 (0.92–1.54), 0.18 | 1.14 (0.8–1.63), 0.46 |
| **Physical activity** | | | | | | |
| Not at all to <1/week (ref) | | | | | | |
| 1–3 times/week | **0.73 (0.56–0.94), 0.02** | **0.59 (0.47–0.74), <0.001** | 0.93 (0.74–1.16), 0.52 | **0.72 (0.59–0.89), 0.01** | **0.78 (0.68–0.89), <0.001** | **0.52 (0.43–0.62), <0.001** |
| ≥ 4 times/week | **0.60 (0.46–0.80), 0.01** | **0.55 (0.43–0.70), <0.001** | **0.67 (0.52–0.86), 0.01** | **0.71 (0.57–0.88), 0.01** | **0.58 (0.50–0.68), <0.001** | **0.34 (0.27–0.41), <0.001** |

Abbreviations: aOR, Adjusted Odds Ratio; ref, reference. Values in bold are statistically significant. All models (Models 1 to 6) were adjusted for age, gender, civil status, education, household yearly disposable income, labour force status, indigenous status, location, smoking status, alcohol consumption, and physical activity. Values in bold are statistically significant.

type 2 diabetes were 17.61 (OR: 17.61, 95% CI: 10.49–29.54), and 9.55 (OR: 9.55, 95% CI: 5.69–16.03) times higher among obese female and male adults, respectively, compared to their healthy-weight counterparts. Besides, the results showed that obesity is associated with a higher incidence of asthma (OR: 2.64, 95% CI: 1.84–3.80) among female adults, but not statistically significant in the case of male adults (Table 4).

## Discussion

The current study is one of the first pieces of evidence that examined the between-person differences in the association between obesity and common chronic diseases among middle-aged and older Australian adults by utilising three waves spanning nine years of a nationally representative longitudinal survey. After controlling for socio-demographic and behavioural covariates, the longitudinal random-effect logistic regression results reveal that obesity is a major risk factor for chronic diseases (type 2 diabetes, heart disease, asthma, arthritis, and depression).

**Table 4. Adjusted random-effect regression results for the between-person differences in chronic conditions due to obesity stratified by gender.**

| Variables | Model 1 | Model 2 | Model 3 | Model 4 | Model 5 | Model 6 |
| --- | --- | --- | --- | --- | --- | --- |
| | Type 2 diabetes | Heart disease | Asthma | Cancer | Arthritis | Depression |
| | aOR (95% CI) | aOR (95% CI) | aOR (95% CI) | aOR (95% CI) | aOR (95% CI) | aOR (95% CI) |
| **Gender: Male** | | | | | | |
| **BMI Categories** | | | | | | |
| Underweight | 0.46 (0.03–2.61), 0.27 | 2.11 (0.57–7.78), 0.26 | 0.19 (0.02–1.57), 0.12 | 0.53 (0.13–2.18), 0.38 | 1.22 (0.49–3.09), 0.66 | 2.15 (0.64–7.19), 0.21 |
| Healthy weight (ref) | | | | | | |
| Overweight | **3.01 (1.88–4.81), <0.001** | 1.22 (0.85–1.75), 0.27 | 0.79 (0.53–1.16), 0.23 | 0.89 (0.66–1.20), 0.45 | **1.34 (1.07–1.69), 0.01** | 1.05 (0.75–1.47), 0.78 |
| Obesity (≥30) | **9.55 (5.69–16.03), <0.001** | **2.19 (1.44–3.33), <0.001** | 1.17 (0.75–1.84), 0.45 | 0.92 (0.64–1.31), 0.63 | **2.24 (1.71–2.93), <0.001** | **1.96 (1.34–2.87), 0.01** |
| **Gender: Female** | | | | | | |
| **BMI Categories** | | | | | | |
| Underweight | 0.29 (0.04–4.75), 0.51 | **3.43 (1.42–8.25), 0.01** | 0.70 (0.24–2.01), 0.50 | 1.92 (0.90–4.09), 0.09 | 0.99 (0.56–1.77), 0.99 | 1.33 (0.64–2.76), 0.44 |
| Healthy weight (ref) | | | | | | |
| Overweight | **5.02 (3.04–8.28), <0.001** | **1.60 (1.11–2.31), 0.01** | **1.58 (1.14–2.20), 0.01** | 0.76 (0.56–1.04), 0.09 | **1.46 (1.20–1.78), <0.001** | **1.43 (1.10–1.87), 0.01** |
| Obesity (≥30) | **17.61 (10.49–29.54), <0.001** | **1.83 (1.22–2.73), 0.01** | **2.64 (1.84–3.80), <0.001** | 0.89 (0.64–1.24), 0.50 | **2.25 (1.80–2.81), <0.001** | **1.96 (1.46–2.62), <0.001** |

Abbreviations: aOR, Adjusted Odds Ratio; ref, reference. All models (Models 1 to 6) were adjusted for age, gender, civil status, education, household yearly disposable income, labour force status, indigenous status, location, smoking status, alcohol consumption, and physical activity. Values in bold are statistically significant.

This study identified obesity as a significant risk factor for type 2 diabetes. This notion fits well with previous findings [15, 24], wherein the authors concluded that overeating and obesity were strongly associated with type 2 diabetes. The present analysis has also revealed a significant positive relationship between obesity and the risk of heart disease. Identical results are available in numerous past studies showing that increasing BMI increases the risk of heart failure in both men and women [25]. Excess weight is a high-risk factor for ischemic stroke and hemorrhagic stroke [26]. A recent study demonstrated that the increased risk of heart disease might be due to a higher incidence of hypertension, adverse hemodynamic effects, maladaptive modifications in cardiovascular structure and function and increased atrial fibrillation among obese people [27].

The finding of a positive association between obesity and asthma is consistent with the existing literature [28]. The possible reason could be that obesity affects lung function by superfluous tissues constricting the thoracic cage, increasing the chest wall's insinuation with fat tissue and pulmonary blood volume [29]. Besides, obesity also causes changes in lung volume and respiratory muscle function [30], leading to asthmatic problems.

Another novel finding of the present study is that obesity is a statistically significant risk factor for arthritis in Australian adults. Other studies estimating the association indicated that obesity is a major risk factor of osteoarthritis for Australian adults [31], and there is evidence that a 5-unit in BMI increases the risk of osteoarthritis (knee) by 35% [32]. The possible reason might be obesity causes increased pressure on the knee joints during daily activities, which causes proliferation of periarticular bone, leading to decreased joint space [33].

The present study findings reveal that obese adults are more likely to develop depression irrespective of socioeconomic and demographic status. Many studies have come to identical

conclusions [16, 34, 35]. There are several reasons for this association. Obese and overweight people generally have low health status and higher co-morbidities (severe chronic diseases) which might cause depression [34]. Apart from this, a model developed by Markowitz et al. illustrated that lack of mobility, lower quality of life and physical functionalities, social stigma and dissatisfaction with body size caused by overweight and obesity, contributes to a higher level of depression [36]. The systematic literature review of Preiss et al. [16] identified eating disorders, interpersonal effectiveness and experience of stigma as other key factors influencing the relationship between co-morbid obesity and depression.

Interestingly, this study observed no significant association between obesity and cancer among adults in Australia. The findings are contradictory to some of the existing literature. In an earlier review, Calle et al. commented that obesity increases the risk of selected types of cancer [37]. Renehan et al. conducted a meta-analysis on BMI and cancer incidence, and they found that obesity is a significant risk factor for developing cancer, and the association was consistent in several continents of the world [38]. Besides, several other studies concluded that obesity-related biological mechanisms (e.g. hormones, calorie constraints, growth factors, inflammatory progressions) influence the development of malignant cells in the body [39, 40]. Therefore, the findings of the lack of association in our study should be interpreted with caution. It should be noted that the HILDA survey does not specify which type of cancer the respondents have developed. Hence, one possibility is that the most common type of cancers (e.g. skin, prostate, colorectal, melanoma and lung) associated with Australian adults are insignificantly impacted by obesity and overweight. Future research should focus on addressing this issue.

Finally, similar to the common knowledge in the public health literature, the results indicate that increased physical activities reduce the risk of chronic diseases irrespective of obesity and socio-demographic status. Noticeably, the most considerable positive impact of physical activities was on the level of depression. Participants engaged in physical activities more than three times a week had a 40% less probability of suffering from chronic depression than those that did not undertake physical activities. An extensive literature related to Australian adults validates this study finding [41, 42]. Therefore, the present study suggests the promotion of physical activities to prevent chronic diseases in Australian adults. The study's findings suggest that physical activities, community-level gym facilities, and the availability of nutritionists to curb excessive weight are necessary. This study calls for future research that will explore the potential of lifestyle interventions and dietary modification to curb excessive weight gain.

Managing obesity has the potential to reduce the prevalence of and mortality from these chronic diseases [43], and improve health-related quality of life [44]. A previous study has claimed that the prevalence of diabetes, high cholesterol, high blood pressure, and CVD among Australian adults could be reduced significantly by reducing body weight [12]. Policymakers and health practitioners might use these findings to devise appropriate strategies and targeted health programs for overweight and obese Australians to reduce their probable burden of chronic diseases.

## Conclusion

This study explores the longitudinal association between obesity and chronic diseases in Australian adults. The longitudinal random-effect logistic regression results showed significant associations between excess body fat (obesity) and chronic diseases. Association between obesity and chronic diseases using longitudinal data is relatively uncommon. This study is one of the few studies that considered six different types of chronic conditions covering nine years of data. The study found that the prevalence and incidence of chronic conditions, such as type 2

diabetes, heart diseases, asthma, arthritis and depression, are higher among obese adults than their healthy-weight counterparts. More specifically, people with obesity are at higher risk of having type 2 diabetes (compared to their healthy counterparts) than any other chronic disease in Australia. The present study has several strengths. Firstly, this study identified which chronic diseases have the strongest association with obesity in Australian adults. Secondly, this study considered a wide range of chronic diseases while checking their relationship with obesity. Thirdly, unlike previous studies, this study employed longitudinal data from the HILDA survey, which is broadly representative of the national population. Fourthly, this study has identified that obesity increase the incidence of chronic diseases differently among men and women.

This study has some drawbacks in estimating the relationships between obesity and chronic diseases. Firstly, this study used self-reported data on BMI, chronic diseases, and lifestyle characteristics. Secondly, this study formed an unbalanced panel data for the subsample analyses. Therefore, causality cannot be drawn from the present study findings. Thirdly, this study did not consider genetic or familial aggregation factors, which are common causes of some chronic diseases, such as type 2 diabetes. Fourthly, the HILDA survey questionnaire does not specify the exact type of cancer or arthritis the participants have developed.

## Acknowledgments

The authors would like to thank the Melbourne Institute of Applied Economic and Social Research for providing the HILDA data set. This paper uses unit record data from the Household, Income and Labour Dynamics in Australia Survey (HILDA) conducted by the Australian Government Department of Social Services (DSS). The findings and views reported in this paper, however, are those of the authors and should not be attributed to the Australian Government, DSS, or any of DSS contractors or partners. DOI: 10.26193/OFRKRH, ADA Dataverse, V2".

## Author Contributions

**Conceptualization:** Syed Afroz Keramat, Rezwanul Hasan Rana.

**Data curation:** Fariha Farjana.

**Methodology:** Rupok Chowdhury, Rubayyat Hashmi.

**Software:** Syed Afroz Keramat, Rubayyat Hashmi.

**Supervision:** Khorshed Alam, Jeff Gow, Stuart J. H. Biddle.

**Writing – original draft:** Syed Afroz Keramat, Rezwanul Hasan Rana, Rupok Chowdhury, Fariha Farjana.

**Writing – review & editing:** Khorshed Alam, Rubayyat Hashmi, Jeff Gow, Stuart J. H. Biddle.

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
