## [Decision Letter · Decision Letter 0]

9 Jun 2021

PONE-D-21-14337

Obesity and chronic disease burden in Australia: Findings from a longitudinal population survey, 2009-2017

PLOS ONE

Dear Dr. Keramat,

Thank you for submitting your manuscript to PLOS ONE. After careful consideration, we feel that it has merit but does not fully meet PLOS ONE’s publication criteria as it currently stands. Therefore, we invite you to submit a revised version of the manuscript that addresses the points raised during the review process.

We look forward to receiving your revised manuscript.

Kind regards,

David Meyre

Academic Editor

PLOS ONE

Journal Requirements:

"NO. The funders had no role in study design, data collection and analysis, decision to publish, or preparation of the manuscript."

Reviewers' comments:

Reviewer's Responses to Questions

**Comments to the Author**

1. Is the manuscript technically sound, and do the data support the conclusions?

Reviewer #1: No

Reviewer #2: Partly

Reviewer #3: Yes

2. Has the statistical analysis been performed appropriately and rigorously? 

Reviewer #1: No

Reviewer #2: No

Reviewer #3: Yes

3. Have the authors made all data underlying the findings in their manuscript fully available?

Reviewer #1: Yes

Reviewer #2: No

Reviewer #3: No

4. Is the manuscript presented in an intelligible fashion and written in standard English?

Reviewer #1: No

Reviewer #2: No

Reviewer #3: Yes

5. Review Comments to the Author

Reviewer #1: This is an important topic. Obesity and non-communicable diseases are both correlated and with a pathophysiological explanation and well-established theoretical framework behind their link. However, there isn’t enough high-quality temporal evidence from Australia, and this study utilizes longitudinal data from an Australian questionnaire to further provide a broader perspective.

My major concern in this article is that it opts to establish associations for a very wide age range between BMI and NCD. Clinically and public health-wise, the interpretation of findings critically depends by age and it is certainly not something that a simple adjustment to age solves. This holds for all study outcomes, and especially to diabetes and cancer. Therefore, stratification of all data to age range is critical here. This is specifically relevant to cancer where there is no cancer-specific data. In this case working with more homogenous groups age-wise is important for mitigating cancer case-mix. The fact that overweight was nearly protective is this cohort OR=0.82 (upper CI, 1.02) is a good evidence for that. A recent Lancet DE paper demonstrated the complexity of looking on the entire cancer types together given critical sex-related differences (32027851). With this respect reverse causality is a serious source of bias that may not have received the attention it deserves in the discussion and analytically. (for example, consider applying a wash out period [one wave]).

In general, presentation of critical data is missing. For example the number of cases diagnosed in each wave, including their reported BMI. Not really clear how people who switched categories were handled. More is needed to be added to the Methods to clarify these.

Other comments:

• Abstract

• Conclusion, line 25: "could be helpful" instead of "would be helpful".

• Introduction:

• Missing reference to the World Health Organization data on obesity (line 4)

• Line 18: should say "it is strongly associated with a higher incidence of type 2 diabetes and depression".

• Page 4, line 1: "…compared to men" – should explain why normal men? Is there literature about men suffering from less obesity-related diseases? This is what's inferred from this sentence (and is not true according to the results of this study).

• Page 4, line 4: perhaps replace "dynamic" with "temporal".

• Theoretical framework

• Page 5, figure 1: should say "insulin resistance" instead of "resistant".

• Page 5, line 1: abdominal obesity is correlated with asthma, it does not lead to asthma.

• Exposure variable: the exposure variable is BMI, however some of the study subjects are 15-19 years old. Adult BMI values are only relevant from ages 20 and above. Ages 2-19 have a different way of calculating their BMI percentile. Study should account for these differences and explain it in this section.

• Outcome variable: it is unclear if patients from the first wave (wave 9) are excluded if they suffer from the disease (exclusion criteria seems to be pregnant women only). If not, obesity is found to be prevalent with the disease, but one cannot conclude that it is a risk factor.

• Results

• figure 3: it is unclear if this figure illustrates the prevalence of chronic diseases among obese people, or the occurrence.

• Table 3: In general, most of the data is somewhat distracting and I suggest you move it to the supplements. it is also unclear if these odds ratio are adjusted to certain confounders. If so, to which ones? If not – why not? It is shown in table 2 that gender, for example, is a confounder in regard to obesity in this study sample.

• Another table of different categorical BMIs in the first wave (wave 9) with their diseases in the last wave (wave 17) could be added (accounting for those who were already sick in wave 9).

• Discussion

• Page 14, line 6 – "cancer" is mentioned as a major risk factor for chronic diseases, even though it is not shown in this study.

• Could there be an explanation to the fact that this study is supposed to be representative of the general Australian population, but 58% are overweight/obese and the introduction says 67% are? That's almost a 10% difference. A volunteer selection bias is suspected.

• The subsample analyses should be further explained, as it is described as a study limitation, but the method of analysis was not mentioned in the article. It is unclear why causality could not be drawn from the study findings.

Reviewer #2: I commend the authors for their approach to studying obesity and the risk of different chronic diseases in the Australian population. The HILDA dataset used seems like a great choice for this analysis. However, I found that the paper was lacking in a lot of details about how the analyses was conducted. Substantial revisions are required for full transparency of the methods. Specifically, I had trouble understanding if the data was cross-sectional or longitudinal. The authors repeatedly stated that the data was longitudinal, but often the presentation of the data made it appear cross-sectional. This needs to be sorted out. The longitudinal nature of the data is the novel part of this project. 

I also question the selection of participants for analyses. The authors chose to include all participants aged 15 and over in their study. However, this is problematic given that the chronic diseases being investigated are very different in younger individuals than in older adults. This can be observed in the main analyses which shows astoundingly high odds ratios for age with most of the conditions. This finding makes sense, however it skews the interpretation of the results. The results are an average across all the age groups adjusted for, and I'm not convinced that the associations of the other variables including BMI category with the outcomes should be pooled across age groups. The analyses should either be stratified by age, or it should be limited to older adults only. The authors should also consider conducting sex-stratified analyses because the risk factors for the outcomes can also substantially differ between males and females. 

For transparency, I have not reviewed the discussion, but am happy to do so after the analyses has appropriately been revised. I feel that I am unable to provide meaningful comments on the discussion when I am very uncertain about the validity of the results.

Page 3, Line 3 - rate of obesity or prevalence? Rate implies a unit of measurement over time. 

Page 3, Line 4 - Why not just write "It is also estimated that at least 7% of deaths from all causes globally"?

Page 3, Line 5 - Not sure predominance is the word you're looking for - prevalence? 

Page 3, Line 9 - In general this first paragraph could be more concise. Overweight and obesity are prevalent in Australia and globally, with the prevalence currently increasing. This is well known information and can therefore be briefly mentioned. Instead, I would expand upon the point you're making in line 11 - burden of what diseases? Which other risk factors? 

Page 3, Line 12 - I don't think there's much of a debate anymore about the role of obesity in many of the non-communicable diseases you've listed?  

Page 3, Line 17 - Citation? Is the evidence actually robust that obesity increases the risk of heart disease-related morbidity and mortality? Isn't this what people refer to as the obesity paradox (though please note, the obesity paradox is likely not true, but instead is at least partially explained by collider stratification bias)? Given the amount of discussion around the obesity paradox in the literature, I would be cautious with this point. 

Page 3, Line 27 - This paragraph would overall benefit from a better synthesis of evidence. It's choppy and the point you're trying to make gets a little lost

Page 3, LIne 31 - I'm confused about this paragraph - you don't mention anything in your abstract about how SES and demographics, age, life-style behaviours etc. in your abstract. 

Page 4, Line 4 - Are you considering variables like SES and demographics as confounders which should be controlled for or as stratification factors which the first part of this paragraph indicates? 

Page 4, Line 7 - Are you speaking about just the Australian context or internationally? There are quite a lot of nationally representative longitudinal survey out there. 

Page 4, Line 11 - I think the justification could be improved upon. What gaps are are you actually filling with a longitudinal study design? Longitudinal study designs themselves don't fix certain methodological challenges. Rather, how you use longitudinal data can resolve methodological challenges. Your objective should reflect what is actually novel about your project. 

Page 4, Line 18 - This whole theoretical framework belongs in the actual background after making it far more concise. This paper does not specifically look at the mediation of obesity through metabolic conditions to the outcomes of interest, therefore it doesn't make sense to have the theoretical framework be front and center. I'm also not aware of inflammation being considered a metabolic syndrome criteria. Inflammation is certainly an important link between obesity (specifically abdominal obesity which you haven't mentioned) and disease, but it's not usually thought of under the heading of metabolism to my knowledge. 

Page 6, Line 13 - as written, the question seems to be an open text question asking about any illness rather individual questions about specific illnesses. Is that the case? 

Page 6, Line 14 - I'm not sure it's fair to say that classifying individuals based on the WHO categories means you can assess how and in what context pre-obese and obese participants are suspectible to different chronic diseases. Please check if this wording is correct. 

Page 6, Line 24 - Please define "defacto". This is not a term I am familiar with. 

Page 6, Line 24 - What are equivalised household incomes? 

Page 6, Line 26 - Indigenous should be capitalized. Can you please also verify that there are not any rules governing the use of Aboriginal data in Australia. I'm unfamiliar with Australian guidelines, but I know that for many Canadian studies, we are not allowed to specifically report on Indigeuous People without additional permissions. Given the similar histories of colonization in Canada and Australia, please confirm that there aren't any issues related to this when using HILDA data.  

Page 6, Line 29 - Are there citations available for how you chose to categorize your behavioural risk factor variables? 

Page 7, Line 7 - What do you mean by repeated observations on the same individual were used for subsample analyses?

Page 7, LIne 11 - Not every variable in your model is a confounder, are these covariates? Confounder has a specific meaning statistically. 

Page 7, Line 11 - What does "long-term association" mean? Though there aren't any hard and fast rules about model building, generally it's recommended that a p value of <0.25 for the Wald statistic is considered as a threshold for inclusion. There may be variables that while not significant based on the univariate statistics, may still be important confounders. Using p<0.25 helps ensure these are included. What techniques did you use for model building and deciding which variables were actually relevant? I would recommend following the guidance of  Hosmer DW, Lemeshow S. Applied Logistic Regression, 2nd ed. John Wiley & Sons, Inc: New York, New York; 2000.

Page 7, Line 15 - For GEE models, the inferences are only valid when data is missing completely at random which is rarely true for observational studies. If data is missing at random, additional considerations such as using the inverse-probability-weighted method are required. Please comment about how you handled missing data (based on your methods, there could be missing data for some covariates as they were not an inclusion/exclusion criteria). Also please go back to Page 5, Line 26 and provide a flow of participants. The 41,169 person-year observations from 20,145 participants were reduced from how many after applying the inclusion and exclusion criteria? 

Page 8 - Consider how many decimal places are actually required. Generally one is sufficient and makes for a far cleaner table. How did you calculate the confidence intervals? Also, this is showing the person-years not the number of people based on the denominator? That is not a good way of presenting data. You can show the baseline characteristics of people at one time point. 

Page 9, Line 4 - I'm having trouble understanding these results. This is just prevalence data? Are the people the same age at all three time points or are people supposed to be aging eight years as time goes on? If it's a series of cross-sectional data measures, that should be made clear. If it's the same individuals at three time points, I worry there may be something wrong. Teh prevalence of things like heart disease or ever being diagnosed with cancer go up with age to a far greater extent than is shown here. 

Page 10, Line 3 - by weight category, not just in obese people. 

Page 11, Line 4 - I'm still confused if this is meant to be cross-sectional or longitudinal. 

Page 12 - Your age categories are problematic. You have hugely inflated odds ratios in the older age groups because you're comparing them to people so young that heart disease etc. are hardly ever diagnosed in. This has major implications for your whole model - your odds ratios for the over variables are averaged over all the age groups. Given that the youngest two age groups are so different from the oldest two, it's hard to understand what the rest of your models actually mean. Either limit your analyses by age or conduct age-stratified analyses. Also, how did you have data for something like civil status and education for the younger individuals? Even a variable like household income has a different meaning in younger versus older adults.

Reviewer #3: This study by Keramat et al. explores the longitudinal association between obesity and 6 chronic diseases in Australian adults. The study utilizes nine years of longitudinal data obtained through a national survey. Overall, the study is well designed, method is appropriate, and the results are clear and well organized. The manuscript is presented in a logical sequence and is easy to understand. The study fills an important gap in literature and the use of nationally representative longitudinal data further contributes to the generalizability of these results to the larger Australian population. Some comments/suggestions are as follows:

1. In the methods section, the explanation for why participants were selected from only 3 waves of the survey is unclear. While the authors mention that “the particular reason behind considering these waves was that these three waves substantially capture the health and lifestyle-related characteristics of the respondents,” this explanation is vague and unclear. If the authors are specifically alluding to the number of survey respondents being the greatest across the three selected waves, then that should be specifically mentioned. Alternatively, if there is another reason or selection criteria that the authors used, then that should be specifically described in this section to make it more clear.

2. In the results section (page 10), line 3 states “Figure 3 illustrates the occurrence of chronic diseases among obese people.” Since the figure depicts the prevalence of disease across all weight statuses and not only the obese group, it may be more accurate to modify this sentence so that it says “Figure 3 illustrates the occurrence of chronic diseases by weight status.” This would also be consistent with Figure heading which states the same.

3. In the results section on page 10, line 12 states “The prevalence of these three chronic diseases were significantly higher in obese females compared with obese male.” The use of the word significant would usually imply a statistically important difference; however, in this case it does not seem like the gender distribution findings presented in table 2 were statistically tested. The authors may consider performing a statistical test (e.g. Chi-square test) to check if the prevalence of the those chronic diseases is in fact significantly higher in obese females compared to obese males.

4. In the footnotes for Table 3, the authors should consider including an indication of all the covariates that were adjusted for in the statistical model. While the authors provide a discussion of covariates in the methods section, including additional footnotes with table 3 in the results section which precisely outline the adjustment factors will be helpful for the readers.

5. Based on the description of covariates in the methods section, it seems like while sociodemographic and behavioral factors were adjusted for as covariates, the authors may not have considered adjusting for the presence of other chronic conditions when investigating the association between obesity and any one particular chronic disease (i.e. while the chronic diseases have only been considered as outcome variables in this case, they can also be potentially considered as covariates). In this case, it would be interesting to see if the association between obesity and each of the chronic diseases remains the same after adjustment for other chronic diseases that are also potential risk factors. For example, it would be of interesting to know whether the association between say obesity and cancer changes when adjusted for other chronic diseases like asthma or diabetes which are also known risk factors for cancer. Adjusting for the other chronic diseases within this context can also allow the authors to potentially explore mediation effects.

6. The sample size of the study should also be indicated in the abstract. While the authors mention using data from three waves of the survey, there is no indication of study size in the abstract.

7. The manuscript requires minor grammatical and syntax revisions. Just to give an example, the following line from page 14 (line 4) contains an error and is unclear: “the multivariate regression results reveal that with longitudinal data signified obesity as a major risk factor for chronic diseases…” It is recommended that the manuscript be revised for all such errors.

6. PLOS authors have the option to publish the peer review history of their article (what does this mean?). If published, this will include your full peer review and any attached files.

Reviewer #1: No

Reviewer #2: No

Reviewer #3: No

---

## [Editor Report · Decision Letter 1]

4 Nov 2021

Obesity and the risk of developing chronic diseases in middle-aged and older adults: Findings from an Australian longitudinal population survey, 2009-2017

PONE-D-21-14337R1

Dear Dr. Keramat,

We’re pleased to inform you that your manuscript has been judged scientifically suitable for publication and will be formally accepted for publication once it meets all outstanding technical requirements.

Kind regards,

David Meyre

Academic Editor

PLOS ONE
---

## [Editor Report · Acceptance letter]

8 Nov 2021

PONE-D-21-14337R1 

Obesity and the risk of developing chronic diseases in middle-aged and older adults: Findings from an Australian longitudinal population survey, 2009-2017 

Dear Dr. Keramat:

I'm pleased to inform you that your manuscript has been deemed suitable for publication in PLOS ONE. Congratulations! Your manuscript is now with our production department. 

Kind regards, 

on behalf of

Dr. David Meyre 

Academic Editor

PLOS ONE